How does abstract and concrete garbage classification signage influence waste sorting behavior?

Cao Gai
Cao Rong caorong@nwu.edu.cn
Liu Peng liupeng@nwu.edu.cn
School of Public Administration/School of Emergency Management, Northwest University , Xi’an, Shaanxi , China
Prpic Valter
Electronic publication date: 2023 Dec 5
Publication date: 2023
Volume: 11
Electronic Location ID: e16597
Received 2023 Jul 20; Accepted 2023 Nov 14
Copyright: © 2023 Cao et al.
Copyright year: 2023
Copyright holder: Cao et al.
License: This is an open access article distributed under the terms of the Creative Commons Attribution License, which permits unrestricted use, distribution, reproduction and adaptation in any medium and for any purpose provided that it is properly attributed. For attribution, the original author(s), title, publication source (PeerJ) and either DOI or URL of the article must be cited.
License URL: https://creativecommons.org/licenses/by/4.0/

Keywords: Nudge, Garbage classification signage, Garbage classification behavior, The concreteness effect, Behavioral experiment

Funding: National Natural Science Foundation of China 72271197 This work was supported by the National Natural Science Foundation of China under grant number 72271197. The funders had no role in study design, data collection and analysis, decision to publish, or preparation of the manuscript.

==============================
Despite the issuance of standardized garbage classification signage, the rate of garbage classification in China remains low. We conducted a pair of laboratory experiments to explore the cognitive processing differences between abstract (including recyclables, hazardous garbage, and food signs) and concrete (including paper, plastic, glass, metal, textiles, batteries, household chemicals, tubes, and food signs) classification signs. We tested a nudging strategy to enhance garbage classification behavior. In Experiment 1, we divided garbage classification signs into two conditions: an abstract condition (comprising abstract signs) and a concrete condition (comprising concrete signs). The Go/No Go task was used to simulate garbage classification behavior. Participants were instructed to press a key when the garbage stimulus matched the classification signs (Go condition) and to refrain from pressing the key when there was a mismatch (No Go condition). The results showed that responses under the concrete condition were expedited compared to those under the abstract condition. This suggests that concrete signage requires less cognitive exertion, thereby enhancing the efficiency of waste classification. In Experiment 2, we optimized the existing bin signage, which predominantly featured abstract signs (traditional condition), and transformed it into a bin signage that emphasized concrete classification signs. These concrete signs were strategically positioned on the upper part of the bins to draw attention (nudging condition). The results suggested that the nudging condition required fewer cognitive resources than the traditional condition, which in turn increased the efficiency of processing garbage classification. This study not only validates the effects of concreteness in garbage classification but also provides effective nudge strategies to complement existing garbage classification management policy tools in a realistic Chinese context.

Introduction

In 2019, China standardized nationwide garbage classification signage to eliminate confusion about how to separate garbage and guide people to sort garbage correctly. However, although research in the signage field has found that standardized signs are effective in helping users understand and comply with desired behaviors, the reality is that standardized classification signs are not effective in improving the efficiency of residential garbage classification (Tong, Liu & Liu, 2020; Ben-Bassat & Shinar, 2006).

To improve garbage classification efficiency, it is necessary to consider how to enhance residents’ classification ability when processing standardized garbage classification signage. There are two ways to achieve this: self-enhancement and external empowerment. Self-enhancement encourages people to learn about existing classification standards through knowledge training, thereby improving their knowledge and ability to sort garbage (Espinosa, Wang & de Soto, 2022; Hertwig & Grune-Yanoff, 2017). External empowerment helps people sort in a way that reduces the cognitive load by mastering and conforming to their cognitive preferences (Thaler & Sunstein, 2008). While the former approach is rarely effective in improving garbage classification, the latter approach is effective because people’s daily behavior tends to be dominated by an intuitive thinking system (Kahneman, 2011). People prefer to make quick decisions based on intuition, which mobilizes fewer cognitive resources, rather than using a rational analytical system (Evans, 2008). As a result, people do not have the patience to stop at garbage bins and learn to sort garbage properly. This is why self-enhancement methods are ineffective in real-life garbage classification campaigns. Instead, it is necessary to explore the cognitive processing mechanisms of the existing standardized garbage classification signage to promote classification efficiency in an empowering way that aligns with people’s cognitive preferences.

Due to variations in garbage classification signage, people use different cognitive processing mechanisms for them. The current national standard (CB/T19095) in China includes primary and secondary garbage classification signage. Primary signage is divided into four categories: recyclable, hazardous waste, food waste, and residual waste. It is further divided into eleven secondary signs: paper, glass, metal, plastic, and textiles fall under the recyclable category; tubes, batteries and household chemicals fall under the hazardous waste category; and household food waste, restaurant food waste and other food waste fall under the food waste category. Individuals have varying levels of psychological representation of the information conveyed by this primary and secondary signage. This leads us to consider construal level theory, which posits that individuals have different levels of abstraction (high construal level) or specificity (low construal level) in their psychological representation of information during the encoding process (Trope & Liberman, 2010; Trope & Liberman, 2003). Representations at a high construal level are primarily decontextualized, involving the construction of an abstract conceptualization of information that captures the commonalities among specific entities. For instance, the concept of ‘recyclable’ involved in primary signage is generated by abstracting the commonalities of all concrete items that can be recycled. Conversely, a low construal level representation includes contextualized information that conveys details in a more specific manner (Chen et al., 2022). For example, ‘plastic’ (secondary signage) included in ‘recyclables’ (primary signage) represents more specific and detailed information. Building on this, our study treats primary signage as an abstract concept, while secondary signage is considered a concrete concept.

Currently, the standardization of waste bin signage in China predominantly features abstract classification markers, with concrete classification markers serving as supplementary information. However, numerous experimental studies have revealed disparities in the cognitive processing of abstract and concrete concepts. It has been observed that individuals’ processing speed and efficiency are higher for concrete concepts than for abstract ones, a phenomenon referred to as the ‘concreteness effect’ (Fliessbach et al., 2006; Borghi et al., 2011). For instance, in a recall task, participants were asked to encode and memorize 90 randomly selected concrete words and 90 abstract words. These 180 previously encoded words (old words) were then mixed with another set of 180 unfamiliar words (new words), and the participants had to identify whether the words were new or old. The findings indicated that the reaction time for concrete words in the new/old judgment task was significantly shorter than the reaction time for abstract words, and the accuracy rate for concrete words was higher (Fliessbach et al., 2006). In a word recognition task where concrete and abstract concept words were presented in different sequences, participants had to determine whether the presented string was correct English vocabulary. The results suggested that concrete concept words held more advantages compared to abstract concept words (Tolentino & Tokowicz, 2009). The concreteness effect was also observed in a sentence validation task. In this task, the last word of each sentence was replaced with a specific or abstract word, and participants were required to judge the authenticity of three groups (image generation, semantic decision, and evaluation of surface characteristics). The results demonstrated that the reaction time for concrete words was lower than that for abstract words across all three groups (West & Holcomb, 2000). In light of the findings related to the concreteness effect, there is a need for further exploration at the cognitive level of the existing signage system, which is predominantly characterized by abstract classification markers. Despite the maturity of research on these concepts, there is a lack of in-depth exploration of standardized waste classification signage. Our study seeks to apply the theory of concrete conceptual dominance to the novel context of waste classification. We investigate whether concrete classification signage can enhance classification efficiency compared to the currently dominant abstract classification signage.

In addition to exploring the cognitive processing mechanisms behind current standardized garbage classification signage messages, we consider how to optimize garbage classification signage for empowerment purposes. Recently, “nudge” theory has gained attention from Western governments as an empowerment approach to raise behavioral awareness (Mahmoud, 2016). Nudge theory, as proposed by Richard Thaler, the Nobel Prize winner in economic sciences and the “father of behavioral economics”, guides individuals to make the best decisions voluntarily (Thaler & Sunstein, 2008). It caters to individuals’ inherent cognitive nature without invoking cognitive ability and allows them to choose freely without prohibiting any options or significantly changing economic incentives (Serim, 2018). The nudging strategy has been employed to subtly steer individuals toward correct waste sorting behavior while accounting for their cognitive preferences. For instance, different inlet shapes have been used to guide waste sorting behavior for various types of waste by relying on people’s familiarity and experience. These include a round opening for plastic bottles, a square opening for paper, and a smaller opening for batteries (Keramitsoglou & Tsagarakis, 2018). Social norms have also been leveraged to encourage waste sorting behavior. An example of this is the use of eye images to activate people’s motivation based on social norms and draw their attention toward written waste sorting instructions, thereby enhancing waste sorting behavior (Lotti, Barile & Manfredi, 2023). Moreover, individuals’ guilt about the environmental damage that can result from improper waste disposal can be triggered to promote waste sorting behavior. For instance, the sorting of plastic waste was improved by displaying images of animals trapped in plastic alongside waste sorting guidelines (Luo et al., 2022). Simplification is another crucial aspect of the nudging strategy for waste sorting. Describing specific operating methods can make classification signage easier to understand, thereby promoting correct waste sorting behavior (Rosenthal & Linder, 2021). Signage that included images or icons was found to be more effective in increasing accurate waste sorting behavior than signage with text only (Wu et al., 2018). In summary, existing nudge research has either been combined with signage or has completely altered the design of the original signage to promote waste sorting. For the standardized waste sorting signage in China, it is necessary to make minor adjustments without altering the essence of the original signage to further encourage this behavior. Despite the urgency of waste sorting in China, there is limited literature on nudging waste sorting in the Chinese context. Therefore, it is necessary to optimize signage characteristics based on people’s cognitive processing preferences to facilitate engagement in waste sorting behaviors in a manner that has not yet been attempted in the Chinese context.

The key to signage guidance behavior is to reduce people’s cognitive load to capture their attention (Wogalter & Laughery, 1996). Although most studies use questionnaires to examine the attention-attracting power of garbage classification signage, questionnaires have difficulty quantifying attentional resources (Xiao et al., 2017; Zhang et al., 2019; Fan, Yang & Shen, 2019). In contrast, laboratory experiments can quantify classification effectiveness using indicators such as response time (Linder et al., 2021; Zhang et al., 2016; Wu et al., 2018). Therefore, this study used a psychological experimental paradigm to examine the response speed and correct rates of garbage classification behaviors and quantify the key attentional resources indicator of signage, remedying the shortcomings of previous studies.

In summary, our research has three innovative aspects: (1) from a theoretical perspective, we have verified the concreteness effects in the context of waste sorting for the first time, thereby filling a theoretical gap in the literature on waste sorting in China; (2) in terms of methodology, our use of psychological paradigms to simulate waste sorting behavior offers a novel supplement to previous research methods on waste sorting behavior; (3) at a practical level, our findings hold significant meaning and impact for the applied case of China, providing the government with a novel approach that aligns with people’s cognitive preferences to achieve effective waste sorting at a low cost. We focus on two issues: (1) whether there is a difference in the cognitive processing mechanism between abstract and concrete garbage classification signage and (2) whether, based on the cognitive mechanism, optimizing garbage classification signage by nudging strategies can empower individuals to improve classification efficiency.

To address these issues, we employed a Go/No Go task to simulate garbage classification behavior and explore the differences between concrete and abstract classification signage at the cognitive processing level. Subsequently, we optimized existing garbage classification signage based on these cognitive processing characteristics to enhance classification efficiency in an empowering manner. Specifically, Experiment 1 manipulated the abstractness of the standardized garbage classification signage (CB/T19095) into abstract and concrete signage. The four primary signs were used as conditions for abstract classification signage (such as recyclables), and the secondary signs served as conditions for concrete signage (such as paper). We hypothesized that individuals’ average response speed under the concrete classification signage condition would be faster than under the abstract classification signage condition. Experiment 2 continued with the Go/No Go task from Experiment 1 and redesigned existing garbage bin signage using a nudging strategy. We anticipated that compared to traditional conditions, the new garbage bin signs would automatically draw individuals’ attention to the concrete classification signage and expedite their response in matching the garbage with the bin, thereby enhancing garbage classification efficiency.

Experiment 1 the concreteness effect of garbage classification signage on classification behavior

Experiment 1 was designed to explore the differences in cognitive processing between concrete and abstract classification signs. This was achieved by altering the level of abstractness of the standardized garbage classification signage (CB/T19095) and transforming it into both abstract and concrete signage through a Go/No Go task that simulated garbage classification behavior. The four primary signs were employed as conditions for abstract classification signage (including recyclables, hazardous garbage, and food signs), while the secondary signs served as conditions for concrete signage (including paper, plastic, glass, metal, textiles, batteries, household chemicals, tubes, and food signs). The task required a keystroke response when the garbage stimulus corresponded with the appropriate garbage bin (Go condition), whereas no keystroke was needed when there was no match (No Go condition).

Subjects

The determination of the sample size was based on two aspects. First, the Go/No Go paradigm (a visual cognitive psychological measurement emphasizing speed and accuracy) adopted in this study typically samples 15–30 participants to obtain reliable concentration trend measures (Liu et al., 2018). Second, we conducted sample-size calculations using G*power (Faul et al., 2007). This calculation was based on the within-subject design of repeated measurement variance. The effect size (f) was 0.4, which was converted based on the minimum effect size (ηp2 = 0.14) from previous research results with similar topics (Wu et al., 2018). According to the software calculation results, a minimum of 10 participants was required to achieve 80% power (α = 0.05) to demonstrate the effect. Therefore, at least 10 participants were recruited in each experiment.

Thirty students (16 males) were recruited to participate in the experiment, with ages ranging from 18 to 29 years (with an average age of 21.43 years). All subjects were right-handed with normal or corrected-to-normal visual acuity and had no history of mental illness. The subjects provided written informed consent and were compensated with a gift at the end of the experimental task. The experiment was approved by the Medical Ethics Committee of Northwest University (approval number 220128001).

Experimental materials and apparatus

The stimulus materials for the experiment included pictures of garbage and garbage bins (Fig. 1). The garbage bin pictures included abstract and concrete classification signage from the national standard (CB/T19095). Modifications were made to the pictures as follows: (1) the current study did not subdivide food garbage, even though the national standard (CB/T19095) subdivides it into household food garbage, restaurant food garbage and other food garbage because the three concrete categories correspond to similar garbage stimuli with different application scenarios; (2) in the abstract classification signage, food waste is referred to as ‘residual garbage from the kitchen’ in Chinese according to the national standard (CB/T19095). However, to distinguish between abstract and concrete classification signs, we altered this terminology to ‘food garbage’ in Chinese in the concrete classification signage. This is because the term ‘food garbage’ in Chinese is more concrete and easier to comprehend than ‘residual garbage from the kitchen’; (3) residual garbage was defined as all kinds of garbage except recyclable garbage, hazardous garbage and food garbage. Because the national standard (CB/T19095) did not define the secondary classification signage of residual garbage, Experiment 1 did not include residual garbage because the concrete classification was not clearly defined.

Figure 1 The garbage stimulus pictures (A) and the garbage bin stimulus pictures (B and C) used in Experiment 1.

The three garbage bins with abstract classification signage are recyclable garbage, hazardous garbage and food garbage. Recyclable garbage was divided into plastic, metal, glass, paper and textile concrete garbage. Hazardous garbage was divided into household chemicals, batteries and concrete garbage tubes. Food garbage was called residual garbage from the kitchen in Chinese in the abstract classification signage and changed to food garbage in Chinese in the concrete classification signage.

The screen was positioned at a viewing distance of 60 cm. The images of garbage were displayed with a field of view measuring 10° × 14° of visual angle, while the images of garbage bins were adjusted to a visual angle of 14° × 17°. All images were standardized for brightness and contrast to ensure consistency across conditions.

E-prime 2.0 software (version 2.0; Psychology Software Tools, Inc. Pittsburgh, PA, USA) was used as the stimulation presentation software, and a Lenovo computer was connected to a 19-inch LCD display (60 Hz refresh rate). A standard keyboard recorded the response times (RTs) and correct rates of the behavioral experiment results.

Procedures

The experiment was conducted in a standardized behavioral laboratory. The subjects sat in front of a computer display that was 60 cm from their eyes, and their chins were placed on a support fixed on the table to keep the center of the display screen at the same level as the central line of their eyesight. Before the formal experiment, the subjects first read the garbage classification guide learning materials and then began the main experiment.

All experimental stimuli were presented on a white background. Each trial was initiated with a central fixation cross (300 ms), followed by a blank screen (300 ms). A garbage stimulus picture was then presented for 500 ms, followed by a random blank screen for 50–150 ms. Next, a garbage bin stimulus picture was presented for 1,500 ms (Fig. 2). The trial interval was 1,000 ms. The experiment required the subjects to respond based on the consistency of the garbage stimulus and the garbage bin stimulus following specific rules. If the garbage stimulus matched the garbage bin stimulus type, the subject pressed the space bar on an English keyboard using the right index finger (Go trial). If the garbage stimulus did not match the garbage bin stimulus type, the subject did not press the space bar (No Go trial).

Figure 2 Trial procedure of Experiment 1.

Experiment 1 used a 9 × 2 × 2 within-subjects design and manipulated the types of garbage stimulus (paper, plastic, glass, metal, textiles, batteries, household chemicals, tubes and food garbage), Go/No Go signals (Go and No Go), and the abstractness of the garbage classification signage (abstract classification signage and concrete classification signage). The experiment was divided into a practice task and a formal task. The practice task consisted of nine trials. The formal task could not begin unless the accuracy of the practice stage reached 100%. The abstract classification signage and concrete classification signage tasks were completed in the formal experimental stage, with the order of the two blocks of tasks balanced among the subjects to avoid the sequence effect. Each block contained 324 trials, with the Go and No Go conditions occurring at a ratio of 2:1. There were 216 trials for the Go signal (including nine levels of the target stimulus type ×24 repetitions) and 108 trials for the No Go signal (including nine levels of the target stimulus type ×12 repetitions). The subjects took a break of at least 2 min after every 81 trials and were encouraged to take longer breaks to eliminate visual fatigue.

Results

Since the No Go trials did not require a keystroke response and were only used to match the Go trials rather than to simulate garbage classification behavior, the experimental results only analyzed the RTs and correct rate indexes of the Go trials.

First, a 9 × 2 within-subjects analysis of variance (ANOVA) was conducted on the RTs as a function of the garbage stimulus types (paper, plastic, glass, metal, textiles, batteries, household chemicals, tubes and food garbage) and the abstractness of the garbage classification signage (abstract classification signage and concrete classification signage). The analysis of RTs showed significant main effects of the garbage stimulus type (F (8, 232) = 13.557, p < 0.001, η2p = 0.319) and the abstractness of the garbage classification signage (F (1, 29) = 33.542, p < 0.001, η2p = 0.536), together with a significant two-way interaction between these two factors (F (8, 232) = 13.440, p < 0.001, η2p = 0.317). A paired-sample t test was used to analyze the differences between the abstract and concrete classification signage conditions for the nine garbage types (Table 1). The results indicated that for the garbage categories of paper, glass, metal, textiles, tubes, batteries and household chemicals, the mean RTs in the abstract classification signage condition were significantly slower than those in the concrete classification signage condition (ps ≤ 0.003), while the difference between the abstract and concrete classification signage conditions was not significant for either food or plastic garbage categories (ps ≥ 0.133).

Table 1 Mean RTs and standard deviations (mean ± SD), detailed paired-samples t test results and the 95% confidence intervals between the abstract condition and the concrete condition in Experiment 1.

Garbage	Abstract classification signage (ms)	Concrete classification signage (ms)	t	p	Cohen’s d	95% confidence interval	
Lower	Upper	
Food	540 ± 101	524 ± 97	1.546	0.133	0.282	−5.021	36.108	
Batteries	573 ± 106	493 ± 83	6.545***	<0.001	1.195	54.883	104.778	
Household chemicals	584 ± 111	541 ± 80	3.289**	0.003	0.601	16.197	69.461	
Tubes	607 ± 114	511 ± 87	6.727***	<0.001	1.228	66.820	125.198	
Glass	590 ± 103	521 ± 96	5.207***	<0.001	0.951	41.742	95.753	
Metal	607 ± 113	514 ± 84	6.445***	<0.001	1.177	63.421	122.383	
Paper	548 ± 103	493 ± 84	4.128***	<0.001	0.754	27.907	82.710	
Plastic	570 ± 110	560 ± 95	0.716	0.480	0.131	−18.606	38.645	
Textiles	574 ± 106	496 ± 87	5.886***	<0.001	1.075	51.131	105.587	
Notes:

*** p < 0.001.

** p < 0.01.

A 9 × 2 within-subjects analysis of variance (ANOVA) was conducted on the correct rates as a function of the type of garbage stimulus (paper, plastic, glass, metal, textiles, batteries, household chemicals, tubes and food garbage) and the abstractness of the garbage classification signage (abstract classification signage and concrete classification signage). The results of the correct rates revealed that the main effect of the garbage stimulus type (F (8, 232) = 2.371, p = 0.088, η2p = 0.076) was not significant. However, the main effect of the abstractness of the garbage classification signs (F (1, 29) = 7.856, p = 0.009, η2p = 0.213) and the two-way interaction between the garbage stimulus type and the abstractness of the garbage classification signs (F (8, 232) = 3.011, p = 0.046, η2p = 0.094) reached significance. A paired-samples t test was then used to analyze the difference in correct rates between the abstract and concrete classification signage conditions for the nine types of garbage pictures (Table 2). The correct rate for food garbage in the abstract classification signage condition was significantly higher than in the concrete classification signage condition (p = 0.04). In contrast, the correct rates for tubes and textiles garbage in the concrete classification signage conditions were significantly higher than in the abstract classification signage conditions (ps ≤ 0.044). The correct rates of the remaining six types of garbage were not significantly different between the two conditions (ps ≥ 0.044).

Table 2 Mean correct rates and standard deviations, paired-samples t test results and the 95% confidence intervals between the abstract condition and the concrete condition in Experiment 1.

Garbage	Abstract classification signage (%)	Concrete classification signage (%)	t	p	Cohen’s d	95% confidence interval	
Lower	Upper	
Food	1.00 ± 0.00	0.99 ± 0.01	−2.112*	0.043	0.386	0.0001	0.010	
Batteries	0.99 ± 0.02	1.00 ± 0.00	1.980	0.057	0.361	−0.014	0.0002	
Household chemicals	0.99 ± 0.03	1.00 ± 0.01	1.756	0.090	0.321	−0.020	0.002	
Tubes	0.97 ± 0.08	1.00 ± 0.01	2.107*	0.044	0.385	−0.062	−0.001	
Glass	0.99 ± 0.02	0.99 ± 0.02	−1.000	0.326	0.183	−0.006	0.016	
Metal	0.98 ± 0.02	0.99 ± 0.02	1.306	0.202	0.238	−0.017	0.004	
Paper	0.99 ± 0.02	1.00 ± 0.01	1.140	0.264	0.208	−0.011	0.003	
Plastic	0.99 ± 0.02	1.00 ± 0.01	1.000	0.326	0.183	−0.012	0.004	
Textiles	0.99 ± 0.02	1.00 ± 0.01	2.804**	0.009	0.512	−0.018	−0.003	
Notes:

** p < 0.01.

* p < 0.05.

Additionally, we performed a differential test on three demographic variables (gender, education level, and major) under both the abstract classification signage and concrete classification signage conditions. As shown in the Appendix, the results indicated that gender, education level, and major had no significant impact on different conditions, thus excluding the interference of demographic differences.

Discussion

The results indicated that the mean RTs for paper, glass, metal, textile, battery, household chemical, and tube garbage images were faster in the concrete classification signage condition than in the abstract classification signage condition. This confirmed our hypothesis that concrete signage has a dominant effect on these seven types of garbage stimuli; it consumes fewer cognitive resources than abstract signage and results in faster response times. In contrast, recognizing abstract classification signage requires deeper processing of the categories to which the garbage stimuli belongs, which consumes more cognitive resources and thus slows RTs.

Interestingly, the mean RTs for food and plastic garbage did not show a significant difference between the two conditions. This suggests that individuals process information about food waste similarly under both abstract and concrete signage conditions. This could be attributed to two reasons. First, individuals are more familiar with food waste and believe that all edible substances belong to this category. Therefore, determining whether something is food waste depends more on personal experience than on signage guidance. Second, in this study, food waste consisted of organic substances, which distinguishes it from recyclable and hazardous waste made of inorganic substances. This allowed individuals to differentiate food waste from other types of waste based solely on its organic properties. As a result, the distinction between abstract and concrete classification signage guidelines was less significant for food waste. Similarly, there was no difference in individuals’ cognitive processing of plastic waste between abstract and concrete signage. This suggests that the participants automatically associated plastic waste with recyclable waste, resulting in no significant difference in their cognitive processing speed between the two types of classification signage. This could be attributed to China’s waste recycling mechanism where people commonly collect plastic products such as plastic bottles and exchange them for money at waste recycling stations. This practice has ingrained in participants that plastic bottles are representative of recyclable materials.

The correct rate results showed that the correct rates for tube and textile garbage in the concrete classification signage condition were significantly higher than those in the abstract classification signage condition, which verified the hypothesis that concrete signage is easier to understand. However, for food garbage, the subjects’ correct rate was reversed, which was inconsistent with the hypothesis. Combined with the RT results, it is possible that people had abstract cognitive preferences for food garbage due to experience, which improved the correct rate. In addition, there was no significant difference in the correct rates of the remaining six types of garbage between the two classification conditions. A possible reason is that prior to the formal experiment, to ensure consistency in the subjects’ experience, they were already familiar with the classification rules, resulting in an overall correct rate of over 90% (the ceiling effect). Therefore, Experiment 1 verified the dominant effect of concrete signage and provided a cognitive basis for the nudging design of Experiment 2.

Experiment 2 nudging strategy to improve the garbage classification efficiency

Experiment 2 continued the Go/No Go task to simulate garbage classification behavior, but with a twist: the existing garbage classification signs were redesigned using a nudging strategy. The current bin signage, which predominantly used abstract signage with concrete signage as a supplement (traditional condition), was replaced with a setup where concrete signage was dominant and abstract signage was secondary. Concrete classification signs were strategically placed on the upper part of the garbage bins to emphasize their attentional priority (nudging condition). The objective was to examine the differences between these two conditions. It was anticipated that the response speed in the nudging condition would be faster.

Subjects

To ensure the stability of the results, we recruited thirty-four subjects in Experiment 2. Four subjects, 3, 4, 7, and 19, were excluded from the data analysis. Subject 19 was excluded due to a low correct rate, while subjects 3, 4, and 7 were so familiar with garbage classification that there was no difference in their reaction patterns between the two conditions. The remaining 30 subjects (17 males) were between the ages of 18 and 29 (average age 21.7 years), with one left-handed subject. All subjects had normal or corrected-to-normal visual acuity and no history of mental illness. The same gift as in Experiment 1 was offered at the end of the experimental task. The experiment was approved by the Medical Ethics Committee of Northwest University.

Experimental materials and apparatus

As shown in Fig. 3, the abstract classification signage (recyclable, hazardous garbage, food garbage and residual garbage) was divided into four categories according to the national standard (CB/T19095), with a total of 16 concrete classification signs. The national standard (CB/T19095) originally included five concrete categories of recyclables, but since there was no significant difference in reaction time between the plastic and recyclable categories in Experiment 1, only four concrete categories of recyclables were retained: paper, glass, metal and textiles. The national standard (CB/T19095) included three concrete categories (household chemicals, tubes and batteries), and a category of expired drugs was added to maintain uniformity with the number of concrete categories in the recyclable category. Similarly, food garbage was divided into leftovers, peels, bones and vegetable leaves, and residual garbage was divided into contaminated paper, broken ceramic products, disposable tableware, and cigarette butts. Each concrete category included two kinds of garbage stimulus pictures. There were eight kinds of pictures of the garbage bins. Four were traditional garbage bin signs (traditional condition) in which the abstract classification signs were placed at the upper part of the bins (middle panel of Fig. 3), while the other garbage bins belonged to the nudging condition, with abstract classification signs placed at the bottom part of the bins and four concrete signs arranged in the form of a 2 × 2 matrix placed at the upper part of the bins (Fig. 3C). The garbage and garbage bin pictures were presented on the same screen as in Experiment 1 to avoid the impact of memory. Experiment 2 included 256 trials with 32 garbage pictures × eight garbage bin pictures. The size of the target stimulus images was adjusted to a visual angle of 10° × 19°. All target stimulus images were standardized to ensure that their luminance and contrast were consistent under different conditions.

Figure 3 The 32 garbage pictures (A) and eight garbage bins (B and C) used in Experiment 2.

In the traditional condition, the abstract classification signage, recyclable, hazardous, food and residual garbage were placed at the upper part of the bins, and the concrete classification signs (the four concrete signs on each bin were arranged in the form of a 2 × 2 matrix with a yellow frame around it) were placed at the lower part of the bins. The concrete classification signs of recyclable garbage included textiles, glass, paper and metal; the concrete classification signs of hazardous garbage included expired drugs, household chemicals, batteries and tubes; and the concrete classification signs of food garbage included vegetable leaf bones, leftovers, and peels. The concrete classification signs of residual types of garbage included contaminated paper, broken ceramic products, disposable tableware, and cigarette butts. In the nudging condition, the position of the abstract classification signage and the concrete classification signage was reversed.

Experiment 2 used the same apparatus as Experiment 1.

Procedures

The laboratory setting was the same as in Experiment 1. Subjects first read the garbage classification guidelines and then entered the main experiment. Note that the garbage classification method was presented in the form of pictures, and the content included all the pictures and textual information that appeared throughout the task.

Figure 4 shows the flow chart of the main task. All experimental stimuli were presented on a white background. Each trial was initiated with a random blank screen for 400–450 ms followed by a target picture (1,500 ms). The interval was 1,000 ms. Subjects were asked to respond according to whether the type of garbage in the target stimulus above matched the garbage bin below with the following rules: if they matched, the response was executed by pressing the space bar with the right index finger (Go trial); if they did not match, the subject did not need to press the space bar (No Go trial).

Figure 4 Trial procedure of Experiment 2.

The main task was divided into a practice task and a formal task. The practice task consisted of 16 trials, and the formal task could not begin unless the accuracy of the practice stage reached 100%. The formal task consisted of 512 trials, during which the subjects took a break of at least 2 min after every 128 trials and were encouraged to take longer breaks to eliminate visual fatigue.

Experiment 2 used a 4 × 2 × 2 within-subjects design that included manipulating the garbage stimulus types (recyclables, hazardous garbage, food garbage, and residual garbage), Go/No Go signals (Go and No Go), and nudging strategies (traditional and nudging conditions). The formal task contained 512 trials in which the Go and No Go conditions were presented in a 1:1 ratio, with 256 trials for the Go signal (including four levels of garbage stimulus types × two levels of the nudging strategy × 32 repetitions) and 256 trials for the No Go signal ( including four levels of garbage stimulus types × two levels of the nudging strategy × 32 repetitions).

Results

Similar to Experiment 1, the experimental results were analyzed for RTs and correct rates for Go trials.

First, a 4 × 2 within-subjects analysis of variance (ANOVA) was conducted on the RTs as a function of the garbage stimulus types (recyclables, hazardous garbage, food garbage, and residual garbage) and nudging strategies (traditional and nudging conditions). The main effects of the garbage stimulus type (F (3, 87) = 10.245, p < 0.001, η2p = 0.261) and the nudging strategy (F (1, 29) = 19.944, p < 0.001, η2p = 0.407) were significant, with a significant two-way interaction between the two factors (F (3, 87) = 4.128, p = 0.01, η2p = 0.125). A paired-samples t test was used to analyze the differences between the nudging and traditional conditions for the four garbage stimulus types (Table 3). The mean RTs in the nudging condition were significantly faster than those in the traditional condition (ps ≤ 0.046) for recyclables, hazardous garbage, and residual garbage, while the difference between the nudging and traditional conditions was not significant for food garbage (p = 0.067).

Table 3 Mean RTs and standard deviations (mean ± SD), detailed paired-samples t test results and the 95% confidence intervals of the mean difference in reaction time between the traditional condition and the nudging condition for four garbage stimulus types.

Garbage	Traditional condition (ms)	Nudging condition (ms)	t	p	Cohen’s d	95% confidence interval	
Lower	Upper	
Food garbage	1,021 ± 115	1,004 ± 97	1.902	0.067	0.347	−1.313	36.289	
Hazardous garbage	1,080 ± 107	1,046 ± 86	3.238*	0.003	0.591	12.492	55.323	
Recyclables	1,033 ± 99	1,015 ± 84	2.084*	0.046	0.381	0.336	35.453	
Residual garbage	1,084 ± 96	1,034 ± 85	5.416**	<0.001	0.989	31.167	68.988	
Notes:

** p < 0.01.

* p < 0.05.

For the analysis of correct rates, a 4 × 2 within-subjects analysis of variance (ANOVA) was conducted on the correct rates as a function of the garbage stimulus types (recyclables, hazardous garbage, food garbage, and residual garbage) and nudging strategies (traditional and nudging conditions). There was no main effect of garbage stimulus type (F (3, 87) = 2.331, p = 0.109, η2p = 0.074) or nudging strategy (F (1, 29) = 0.189, p = 0.667, η2p = 0.006) and no significant two-way interaction between the two factors (F (3, 87) = 2.206, p = 0.093, η2p = 0.071). A paired-samples t test was used to analyze the differences between the nudging and traditional conditions for the four garbage stimulus types (Table 4). The correct rates of hazardous garbage in the nudging condition were significantly higher than those in the traditional condition (p = 0.032), while the correct rates of the remaining three garbage types were not significantly different between the two conditions (ps ≥ 0.141).

Table 4 Mean correct rates and standard deviations (mean ± SD), detailed paired-samples t test results and the 95% confidence intervals of the mean difference in correct rates between the traditional condition and the nudging condition for four garbage stim.

Garbage	Traditional condition (ms)	Nudging condition (ms)	t	p	Cohen’s d	95% confidence interval	
Lower	Upper	
Food garbage	0.99 ± 0.02	0.99 ± 0.02	−0.891	0.38	0.163	−0.013	0.005	
Hazardous garbage	0.99 ± 0.02	1.00 ± 0.01	−2.249*	0.032	0.411	−0.013	−0.001	
Recyclables	0.99 ± 0.02	0.98 ± 0.04	1.514	0.141	0.276	−0.004	0.027	
Residual garbage	0.99 ± 0.02	0.98 ± 0.03	0.682	0.501	0.125	−0.006	0.012	
Notes:

* p < 0.05.

To exclude the interference of demographic differences, we conducted a differential test on three demographic variables (gender, education level, and major) under both traditional and nudging conditions. As shown in the Appendix, the results indicate that gender, education level, and major had no significant impact on different conditions, thus excluding the interference of demographic differences.

Discussion

The RT results demonstrated that the average RTs for recyclables, hazardous waste, and residual waste in the nudging condition (new garbage bins with explicit classification signs) were significantly faster than those in the traditional condition (traditional garbage bins with abstract classification signage). These findings support our hypothesis that in the nudging condition, the highlighted explicit classification signage (positioned at the top and enclosed in yellow frames) automatically attracted attention to the specific categories, thus inducing quicker RTs. Conversely, in the traditional condition, individuals required more cognitive resources to process the conceptual information dominated by the abstract classification signage, leading to a slower response time. Similar to the results of Experiment 1, there was no significant difference in the average RTs for food waste between the nudging and traditional conditions. This could be due to two reasons. First, individuals are more familiar with food waste and believe that all edible substances belong to this category. Therefore, determining whether something is food waste depends on personal experience rather than signage guidance, resulting in comparable cognitive resource consumption between the traditional and nudging conditions. Second, in this study, food waste had distinct characteristics compared to recyclables, hazardous waste, and residual waste. It consisted solely of organic matter, while other types of waste contained a mixture of organic and inorganic matter. This distinction allowed individuals to accurately determine the appropriate garbage bin based solely on its organic characteristics. Consequently, there was a minimal difference between abstract and explicit classification signage guidelines.

The correct rate results showed that the correct rate for hazardous garbage in the nudging condition was higher than that in the traditional condition, which is consistent with the hypothesis. This indicates that highlighting concrete classification signage is easier for people to understand. However, there was no significant difference in the correct rate for the remaining three garbage items between the two conditions, which is inconsistent with the hypothesis. The possible reason is similar to Experiment 1. To establish consistency in the subjects’ experience, they were familiar with the classification rules before the formal experiment, resulting in an overall correct rate of more than 90% (the ceiling effect).

General discussion

The current study adopted a Go/No Go task to simulate garbage classification behavior in a laboratory context. Its objectives were twofold: (1) to explore the differences in cognitive processing mechanisms between abstract and concrete garbage classification signage and (2) to investigate whether garbage classification signage optimized by nudging strategies can empower individuals to improve classification efficiency. The goal was to promote garbage classification behavior from an application perspective. In Experiment 1, garbage classification signage was divided into abstract classification sign conditions (recyclables, hazardous garbage and food garbage) and concrete classification sign conditions (paper, plastic, glass, metal, textiles, batteries, household chemicals, tubes and food) to examine the differences between the two conditions. The results showed that the concrete classification signage condition improved the processing efficiency of garbage classification behavior by reducing the consumption of cognitive resources compared to the abstract classification signage condition. Experiment 2 redesigned traditional waste bin signage using a nudging approach. The original design, which primarily featured abstract signage with concrete signage as a supplement (traditional condition), was replaced with a design that emphasized concrete signage as the primary focus and abstract signage as secondary. Specifically, concrete classification signs were positioned at the top of the waste bins to prioritize their visibility (nudging condition). The results showed that the nudging condition consumed fewer cognitive resources than the traditional condition, thereby enhancing the decision processing efficiency of garbage classification behavior in daily life.

This study found that an emphasis on concrete classification signage had a facilitative effect on garbage classification behavior. This is consistent with previous findings on concreteness effects in the conceptual field (Fliessbach et al., 2006; Grootde, Dannenburg & Hellvan, 1994). Previous studies have found that the cognitive processing speed and efficiency for concrete concepts is better than those for abstract concepts in cognitive tasks such as lexical judgment tasks (Ding, Liu & Yang, 2017), recognition tasks (Klaver et al., 2005) and memory tasks (van Schie et al., 2005). Dual coding theory suggests that for conceptual information processing, encoding occurs through verbal and nonverbal (sensory-motor) systems (Paivio, 1991). Concrete concepts are advantageous because they activate both systems, whereas abstract concepts evoke only the verbal system. The availability of multiple processing resources and representations gives concrete words a unique advantage. Brain imaging studies have further shown that cognitive processing of concrete and abstract words is based on partially distinct neuronal mechanisms. The processing of abstract concepts is mainly associated with the involvement of the left temporoparietal and left inferior frontal gyrus, whereas concrete concepts activate both the left and right hemispheres (Fliessbach et al., 2006; Sabsevitz et al., 2005). In conjunction with the present study, we suggest that the lower amount of cognitive resources consumed by concrete signage relative to abstract signage is due to the activation of stronger brain usability representations. Therefore, the concreteness effect is verified in the new context of garbage classification.

The results of this study can also be explained using dual-decision system theory, which includes fast systems (faster intuitive processing) and slow systems (slower rational processing) (Kahneman, 2011). Our existing abstract signage combination model is dominated by abstract signage. When subjects put garbage into garbage bins with abstract classification signage, they need to judge whether the garbage is recyclable garbage, hazardous garbage or food garbage through the slow system (based on the knowledge and experience of abstract garbage classification signage in memory), which consumes a large amount of cognitive resources. In addition, the slow system activated by abstract signage conflicts with the fast system, a permanent mechanism for daily behavior (garbage classification behavior), resulting in slower response times. Comparatively, this study used a nudging approach to visually and quickly match the garbage with the concrete classification signage bins by activating the fast system without extracting it from memory. This not only reduces the consumption of cognitive resources but also responds to the daily behavior mechanism. Therefore, the design of garbage bin should emphasize concreteness to activate fast system decision making and thus improve processing efficiency and nudge garbage classification behavior.

Although studies have found a significant effect of concrete classification knowledge on behavior by using questionnaires, the internal validity of the research conclusions may be insufficient (Barr, Gilg & Ford, 2005; Knussen et al., 2004). Questionnaire indicators mainly examine individuals’ attitudes, which are highly subjective, and questionnaires make it difficult to investigate behavior at the cognitive level (Yla-Mella, Keiski & Pongracz, 2015; Khan, Ahmed & Najmi, 2019; Liu et al., 2015). Previous studies have not quantified the efficacy indicators (attention and comprehension) of garbage classification signage to explore cognitive processes in depth. The current laboratory research method overcomes the shortcomings of questionnaire-based research and complements previous research methods.

Furthermore, the current results can provide cognitive-level evidence for existing theoretical models of garbage classification behavior. The theory of planned behavior (TPB) is the common theoretical basis for current research on garbage classification behavior (Hu et al., 2021; Zhang et al., 2021). The TPB suggests that attitude, subjective norms, and perceived behavioral control are the main factors that affect garbage classification intention and behavior (Ajzen & Madden, 1986; Taylor & Todd, 1995). However, it is arguable how perceived behavioral control affects garbage classification behavior. Perceived behavioral control refers to the individuals’ self-awareness of their knowledge and ability to classify garbage. Some studies have found that perceptual behavior control can directly or indirectly affect garbage classification behavior through intention (Lizin, Van Dael & Van Passel, 2017; Rhodes et al., 2015), while other studies have found that perceptual behavior control does not affect garbage classification behavior (Ramayah, Lee & Lim, 2012). Although the current experiment did not involve a garbage classification intention indicator, the results based on the behavior index suggest that perceptual behavior control can affect garbage classification behavior. Specifically, the abstract and concrete classification signs involved in the current research have different requirements for individual knowledge of garbage classification. Compared to concrete classification signage, the processing of abstract classification signs requires recall of the knowledge and experience of garbage classification through memory. The increase in classification difficulty requires a higher level of perceived behavior control, which reduces decision-making processing efficiency for garbage classification behavior. The findings indicate that perceived behavioral control is an influential factor in garbage classification behavior and provide an experimental basis for the theory of planned behavior.

From a cognitive psychology perspective, this study provides new insights to expand the theory and methodology of research on garbage classification behavior. However, this study does have limitations. First, to ensure consistency in the subjects’ experiences, we allowed them to familiarize themselves with the classification rules in advance. As a result, the overall accuracy rate exceeded 90%. While this made the accuracy index less significant, the response time index remained significant. This reflects the sensitivity of the marker in capturing attention. Therefore, further exploration of the accuracy rate is necessary. Second, it is possible that the procedure used in Experiment 2 facilitated classification simply because the garbage stimuli were closer to the concrete signs in the nudging condition than in the traditional condition. In future research, eye-tracking devices could be used to monitor the eye movements of individuals to further validate this alternative explanation. In addition, although the internal validity of a laboratory task is higher than that of a field experiment, the external validity of the study is insufficient due to differences between laboratory situations and real-life situations. Field experiments or policy applications could yield very different results. In the future, field experiments or policy applications could be adopted to extend the findings of this study to an application level to enhance the practical value of the nudging strategy.

Conclusion and applications

This study arrived at two conclusions through the execution of two experimental tasks. First, the concreteness effect was validated in the novel context of waste sorting. Concrete classification signs reduced the cognitive resources required and expedited individual responses compared to abstract classification signs, thereby enhancing waste sorting efficiency. Second, new waste bins that emphasize concrete classification signs (where the concrete classification signage is positioned at the top of the bins to attract attention) consume fewer cognitive resources during the matching process of waste and waste bins. This provides cognitive evidence to promote waste sorting behavior. The application value of the current study is expanded as follows.

(1) Applying the nudge policy tool promotes garbage classification behavior in China. To improve garbage classification behavior, most current policy tools include the formulation of relevant laws and regulations, economic incentives, education and publicity. However, compulsory laws and regulations, the short-term effect of economic incentives and the cognitive and information overload of education and publicity may be reasons for the low efficiency of garbage classification in practice (Meng et al., 2019; Wang & Hao, 2020). Thus, traditional policy tools encounter a bottleneck in improving garbage classification behavior. As a new policy tool, the nudging strategy has been widely used in research on environmental protection (DesRoches et al., 2023; Kasperbauer, 2017). It aims to automatically guide people to make decisions in a low-cost and nonmandatory way based on their cognitive characteristics, which compensates for the disadvantages of existing garbage classification policy tools. The current study provides new ideas for environmental governance in China.

(2) Exploring individual cognitive psychology and behavior mechanisms is the key to improving the current source management of garbage classification. Individuals’ cognition plays an important role in predicting garbage classification behavior and intention. Cognitive characteristics are an important measure when designing garbage-classification facilities that conform to individual cognitive processing rules. Concrete classification signage can guide individuals to quickly match garbage, thus reducing their cognitive load and improving garbage classification efficiency. Furthermore, this study provides a novel idea and practical method for garbage classification source management from a cognitive perspective.

Supplemental Information

Supplemental Information 1 Analysis of gender, educational level, and major under different conditions in Experiments 1 and 2.

Click here for additional data file.

Supplemental Information 2 Raw data.

Click here for additional data file.

Additional Information and Declarations

Competing Interests

Author Contributions

Human Ethics

Data Availability

The authors declare that they have no competing interests.

Gai Cao conceived and designed the experiments, performed the experiments, analyzed the data, prepared figures and/or tables, authored or reviewed drafts of the article, and approved the final draft.

Rong Cao performed the experiments, authored or reviewed drafts of the article, and approved the final draft.

Peng Liu conceived and designed the experiments, performed the experiments, analyzed the data, prepared figures and/or tables, authored or reviewed drafts of the article, and approved the final draft.

The following information was supplied relating to ethical approvals (i.e., approving body and any reference numbers):

The experiment was approved by the Medical Ethics Committee of Northwest University.

The following information was supplied regarding data availability:

The data is available at ScienceDB: Peng Liu (2022). Nudging Garbage Classification. V1. Science Data Bank. https://doi.org/10.57760/sciencedb.06868.

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
