# Peer review of "How does abstract and concrete garbage classification signage influence waste sorting behavior?"

_PeerJ, doi:10.7717/peerj.16597_

## Round 0.1 · original submission · Major Revisions

Three reviewers found your work interesting and well-written, and I do agree with them. However, they all highlighted different issues that should be addressed before the manuscript is considered for publication. I will not reiterate their comments as they are quite self-explanatory. Please consider all of them and respond point by point in your rebuttal letter.

**Language Note:** The review process has identified that the English language must be improved. PeerJ can provide language editing services - please contact us at copyediting@peerj.com for pricing (be sure to provide your manuscript number and title). Alternatively, you should make your own arrangements to improve the language quality and provide details in your response letter. – PeerJ Staff

Reviewer 1 ·

Basic reporting

The article is written in a clear way, in good English. Literature referencing is overall adquate but I encountered a few concerns which I will address below.

1. The classification of garbage signage as being more or less "abstract" according to construal level theory (Lines 75-79) is not clear enough. Due to this, the definition of "concrete" classification signage is not clear as well. I suggest adding a brief explanation of this theory and explaining in detail how it relates to primary and secondary signage.

2. I suggest to briefly expand the literature reporting about the concreteness effect, e.g.: what were the referenced studies about? which experiments were carried out that led to this theory?

3. I was quite confused to see that much of the methods, hypothesis and even results of the two experiments are anticipated in the introduction section (e.g., line 126: "The task required..."; line 128 "fewer cognitive resources were consumed to process information about concrete classification signage than abstract classification signage, indicating a concreteness effect"). I suggest moving this detailed method's description to the methods section where it belongs and sticking to a brief description of the experiments with relevant hypothesis here.

4. The quality of the figure should be improved. I.e., the font size of labels on the bins in figure 1 is too small and the picture has too low resolution to be zoomed in.

Experimental design

The research question is well defined and linked to the gap in literature it aims to fill. The experiments are conducted to a good technical standard and are described in sufficient detailed.

From times to times the "construal level" of garbage labels is mentioned (e.g., line 161). However, it is not clear what this means. I suggest better addressing this point to aid readers' understanding of your methods.

Validity of the findings

In general, the discussion of the results, which are not univocal but rather heterogeneous, is not very sound. Results that are in line with the main hypotesis are taken as unequivocal evidence that the author's theory is correct and are not further discussed, while those which contradict the hypothesis are easily dismissed (I provide an example below). Therefore, I think this section should be reviewed.

Experiment 1 - The explanation for the lack of significant differences between concrete and abstract labeling for food garbage and plastic does not sound very convincing to me. This point should be further expanded, e.g. what are the organic properties of food and what do they have to do with your experiment? The claim that plastic is easily associated to recyclables does not seem to be supported by any study..?

Additional comments

Here I report a few minor points that are note covered by the three areas above, for the authors to address.

1. From the abstract it is not very clear what the conditions' labels "concrete" and "abstract" mean. I suggest adding an introductory sentence to briefly explain this before jumping to the experiment description.
2. typo at line 54: "gclassification"
3. Line 61-62: I would rephrase as "While the former approach... the latter approach..."
4. Line 74-75: In my opinion the switch from garbage classification to the construal level theory is too sudden. I suggest adding a link sentence to introduce why you are mentioning the theory.
5. Typo at line 181: "Intertial"
6. Line 201: the analysis performed on the RTs (ANOVA) is not mentioned.
7. Typo at line 208: "ps < .003"
8. Line 228: If my understanding of the sentence is correct "However" should be replace with "In contrast"

Reviewer 2 ·

Basic reporting

the title would benefit from being re-worded. English can be improved throughout the paper but it is rarely unclear or hard to follow. literature should definitely be reinforced, there is very little and I struggle to find the references as citations throughout the paper. There are recent papers on the topic, for example Lotti Barile Manfredi 2023, which are not mentioned, and the gaps in the literature are not clearly identified.

Experimental design

The design is overall appropriate, however the authors need to be very clear from the abstract that they run a lab experiment. they also need to reflect on limitations, such as that they run this on computers, that this is a typical case where field experiments or policy applications could give very different results, etc.
Another issue is that in the Experiment 2, upper part and yellow colour are two different nudges, it is difficult to separate the two effects.
I strongly suggest to add a regression with the demographics of participants, if available.

Validity of the findings

The findings are overall very interesting for the applied case of China, but they can also be interesting for the theoretical literature in the field. The study is original but in order to be meaningful and impactful the edits I suggest in the previous section 2 need to be implemented.

Additional comments

It is an interesting experiment overall, there is still some work to do though. Good luck

Reviewer 3 ·

Basic reporting

In two experiments, the authors explored the processing differences between abstract and concrete classification signs and proposed a nudging strategy to improve garbage classification behavior. In the first experiment they compared abstract and concrete signs, in the second one they proposed a new classification sign that highlighted concrete images. In both cases results showed a facilitation of elaboration when concrete information was present or highlighted. The study is interesting and the paper well-written. I have only few minor suggestions for the authors.

Experimental design

1. Can you provide more details about sample size calculation? On which analyses was it based?

2. The issue of the analyses came to my mind when I read the results. If I’m not missing anything, it seems to me that a data analysis section is not written and that some details are missing. I guess the authors used a factorial ANOVA (2 x 2?) for their analyses, but this is not specified. I ask them to report the analyses they conducted either at the beginning of the results section or in a dedicated section “data analyses” at the end of the methods. They also should specify what variables entered into the analyses.

Validity of the findings

How big were the stimuli? Is it possible that the procedure used in Experiment 2 facilitated the classification simply because the garbage stimuli were close to the concrete signs in the nudging condition compared to the control? In absence of eye tracking data that can examine the participants’ gaze behavior, authors should at least acknowledge this potential limitation in the discussion.

---

## Round 0.2 · Minor Revisions

I agree with the reviewer that the manuscript has been highly improved. Please consider the minor requests suggested by the reviewer and proceed with a resubmission.

Reviewer 2 ·

Basic reporting

The authors have put a lot of effort into improving language, literature and certain specific sections that needed edits. There has been an improvement in all aspects.

Experimental design

This component has been refined and it is now more meaningful, appropriate and rigorous.

Validity of the findings

The validity was good already but now new tables discuss the impact of additional and important regressors. I suggest a table with all of them, or at least discuss the results and adding it in the appendix.

Additional comments

I would suggest changing the second "garbage" in the title to "sorting", or something similar? Not to repeat garbage in the title.
Adding the table with all regressors in, these are the 2 small edits I suggest.

---

## Round 0.3 · accepted · Accept

I have assessed the revised version and I can confirm that all the reviewer's comments have been adequately addressed. Therefore, I am happy to accept this manuscript for publication.